# Open-PMC-18M: A High-Fidelity Large Scale Medical Dataset for Multimodal Representation Learning

## Abstract

Compound figures, which are multi-panel composites containing diverse subfigures, are ubiquitous in biomedical literature, yet large-scale subfigure extraction remains largely unaddressed. Prior work on subfigure extraction has been limited in both dataset size and generalizability, leaving a critical open question: *How does high-fidelity image–text alignment via large-scale subfigure extraction impact representation learning in vision-language models?* We address this gap by introducing a scalable subfigure extraction pipeline based on transformer-based object detection, trained on a synthetic corpus of 500,000 compound figures, and achieving state-of-the-art performance on both ImageCLEF 2016 and synthetic benchmarks. Using this pipeline, we release **OPEN-PMC-18M**, a large-scale high quality biomedical vision-language dataset comprising *18 million* clinically relevant subfigure–caption pairs spanning radiology, microscopy, and visible light photography. We train and evaluate vision-language models on our curated datasets and show improved performance across retrieval, zero-shot classification, and robustness benchmarks, outperforming existing baselines. We release our dataset, models, and code to support reproducible benchmarks and further study into biomedical vision-language modeling and representation learning.

## 1 Introduction

The rapid progress of general-domain vision-language models (VLM) (Radford et al., 2021; Jia et al., 2021; Girdhar et al., 2023) has sparked growing interest in building large-scale multimodal datasets tailored to the medical domain (Zhang et al., 2023; Lin et al., 2023a; Pelka et al., 2018; Lozano et al., 2025; Baghbanzadeh et al., 2025). Despite these efforts, the scale of medical datasets still lags far behind their general-domain counterparts. While increasing dataset *size* continues to be a primary goal, there is growing recognition that improving the *quality* and *relevance* of image-text pairs may be a more effective strategy for enhancing model performance and clinical utility (Baghbanzadeh et al., 2025).

Biomedical figures present unique challenges: they often consist of compound layouts that combine multiple subfigures, each potentially depicting a different imaging modality, anatomical region, or clinical concept. Unlike dataset scale, which has received substantial attention, this structural heterogeneity remains largely unexplored. Most of the existing biomedical VLM pipelines treat compound figures as atomic units, pairing the entire image with a caption, without disentangling their internal structure.

We hypothesize that such coarse image-text alignment could introduce noise into pretraining, ultimately impacting the transferability and generalizability of the learned representations. While recent

work has scaled data curation through bulk mining of PubMed Central (PMC)[1] articles (e.g., PMC-15M (Zhang et al., 2023) and BIOMEDICA (Lozano et al., 2025)), these efforts still rely on noisy and compound figures. To our knowledge, only a few prior works incorporate subfigure extraction as part of the curation process (Pelka et al., 2018; Lin et al., 2023a; Baghbanzadeh et al., 2025); however, they do so at small scale. This raises an important gap in the field: *how does subfigure extraction and the resulting improvement in medical image-text alignment quality impact representation learning at scale, particularly given the known sensitivity of contrastive objectives to both dataset size and alignment fidelity during pretraining?*

In this work, we investigate the impact of large-scale subfigure extraction on medical vision-language representation learning. We first create a dataset of 6 million image-caption pairs by filtering out non-medical images (e.g., charts, plots, tables) from the BIOMEDICA corpus (Lozano et al., 2025) using a combination of label metadata and a ResNet classifier. For the *subfigure extraction* step, we train a high-performance object detection model with the same architecture as DAB-DETR (Dynamic Anchor Boxes DEtection TRansformer) (Liu et al., 2022) on a corpus of 500,000 programmatically-created compound figures. By decomposing compound figures with this model, we build OPEN-PMC-18M, one of the largest and most curated collections of biomedical image-text pairs to date, consisting of 18 million subfigure-caption pairs. We then train vision and text encoders using a contrastive learning objective and evaluate the resulting models on an extensive suite of downstream tasks, including cross-modal retrieval and zero-shot classification across three distinct medical modalities: radiology, microscopy, and visible light photography (VLP). We release our dataset[2], models, and code[3] to support reproducible benchmarks and further study into biomedical VLM and representation learning. Our contributions are as follows:

- We propose a scalable subfigure extraction pipeline using transformer-based object detection trained on a 500,000 compound figure dataset, achieving state-of-the-art performance on ImageCLEF 2016 (Kalpathy-Cramer et al., 2014; García Seco de Herrera et al., 2016) and synthetic evaluation sets.

- We release OPEN-PMC-18M, a large-scale biomedical image-text dataset with 18 million subfigure-caption pairs filtered for clinical relevance across radiology, microscopy, and visible light photography.

- We provide a comprehensive evaluation of vision-language models trained on our datasets, demonstrating improved performance in retrieval, classification, and robustness across multiple medical benchmarks.

## 2  Related Work

### 2.1  Biomedical Vision-Language Datasets

Most efforts to date have relied on mining figures and captions from the PMC Open Access subset.[4] One of the earliest publicly available datasets is ROCO (Pelka et al., 2018), which compiled around 80,000 radiology and 6,000 non-radiology images, enriched with metadata such as captions and keywords. Later, Lin et al. (2023b) introduced PMC-OA , which includes 1.6 million image-text pairs. Their contribution emphasized automation—proposing a pipeline to streamline the pairing process and reduce human annotation. More recently, Zhang et al. (2023) announced PMC-15M, a dataset of 15 million image-text pairs. The largest released dataset to date is BIOMEDICA (Lozano et al., 2025), which comprises 24 million pairs and employs clustering, vision encoders, and expert taxonomies to assign modality labels at global and local levels. While these efforts represent major progress in scale, recent work has emphasized that data quality is a critical factor in learning effective and generalizable medical representations (Baghbanzadeh et al., 2025). Building on the premise of OPEN-PMC, our work takes a quality-first approach while also significantly scaling up the dataset.

---

[1]https://pmc.ncbi.nlm.nih.gov/

[2]https://huggingface.co/datasets/vector-institute/open-pmc-18m

[3]https://anonymous.4open.science/r/open-pmc-18m-CE25/

[4]https://pmc.ncbi.nlm.nih.gov/tools/openftlist/

## 2.2 Subfigure Extraction as Object Detection

Early approaches to compound figure separation relied on classical computer vision techniques, using heuristics based on whitespace, edge detection, or layout regularity. However, these methods often struggled to handle diverse panel styles and complex spatial arrangements. More recent work treats subfigure extraction as an object detection problem, leveraging deep learning models. For example, Tsutsui and Crandall (2017) and Yao et al. (2021) used YOLO for subfigure separation. Lin et al. (2023a) also uses an object detection model to extract subfigures in their pipeline. They train a DETR (DEtection TRansformer) (Carion et al., 2020) model on the MedICaT dataset (Subramanian et al., 2020) containing 2069 annotated compound figures.

Data annotation for training an image decomposition model is challenging and time-consuming. Current annotated datasets for this are small, which lead to models with suboptimal performance. To overcome this, synthetic datasets of compound figures have been proposed, where subfigures are programmatically composed to simulate real-world layouts. This allows training of object detection models without relying on large-scale human-annotated data (Tsutsui and Crandall, 2017; Yao et al., 2021).

# 3 Data Composition and Curation Process

## 3.1 Initial Collection and Filtering

We begin with the BIOMEDICA dataset (Lozano et al., 2025), which has been extracted from articles in the PubMed Central Open Access Subset. BIOMEDICA contains approximately 24 million image-caption pairs along with metadata, including global and local modality labels for each image. We apply a filtering step using the provided labels and retain only those pairs primarily categorized as clinical imaging, microscopy, immunoassays, or chemical structure. This yields a dataset of 6 million pairs, which we refer to as PMC-6M in this paper.

## 3.2 Vision-Based Subfigure Extraction

To enable scalable extraction of subfigures from biomedical compound figures, we trained a transformer-based object detection architecture, Dynamic Anchor Box DEtection TRansformer (DAB-DETR) (Liu et al., 2022). Prior work of Lin et al. (2023a) trained a DETR model on MedICaT (Subramanian et al., 2020) with only 2,069 manually annotated compound figures. In contrast, we trained our model on a large-scale synthetic dataset of 500,000 compound figures, the first of its kind in the biomedical domain. We use DAB-DETR as it improves upon the original DETR model by learning dynamic anchors as queries, resulting in improved localization and faster convergence (Liu et al., 2022).

**Synthetic Data Formation.** To train a subfigure extraction model at scale, we generate a synthetic dataset by reversing the subfigure extraction process: rather than decomposing existing compound figures, we programmatically construct new ones by composing multiple single-panel biomedical images into compound layouts. The key advantage of this approach is the availability of ground-truth bounding boxes for each subfigure. Our generation pipeline samples a layout template that specifies the spatial arrangement of subfigures. Each layout is defined by a set of configurable parameters, including:

- **Grid Size**: Specifies a standard $m \times n$ grid or a custom arrangement for panel placement.
- **Margins**: Random horizontal and vertical spacing between panels to simulate variability in published figure layouts.
- **Labeling Scheme**: Determines how panels are annotated (e.g., using numerical, alphabetical, or compound labels like "1a" or "a-1"), and whether labels appear inside or outside panel boundaries.
- **Aspect Ratio**: Specifies a fixed width-to-height ratio applied uniformly to all subfigures.

Subfigures are sampled from a repository of single-panel biomedical images spanning diverse modalities such as radiology, microscopy, pathology, etc., which we will describe below. Composite figures may contain panels from the same modality or a heterogeneous mix, providing semantic

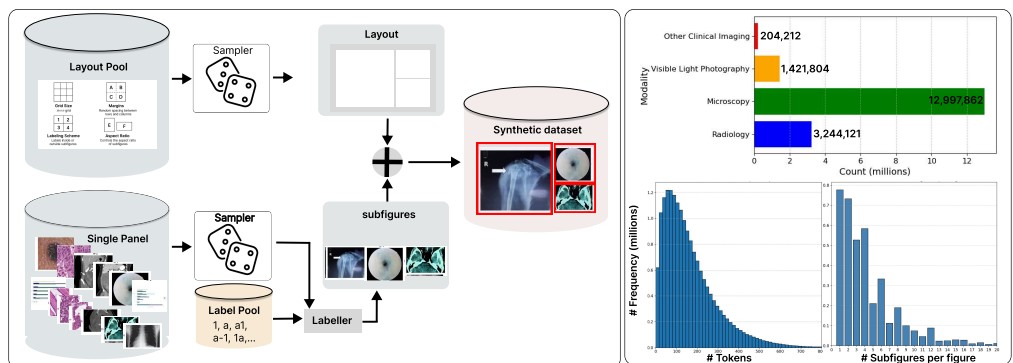

Figure 1: **Left** Overview of our pipeline for creating synthetic compound figures used to train the DAB-DETR model. A *Sampler* selects single-panel images and layout specifications from their respective pools. A *Labeler* assigns subfigure labels from a predefined label pool (e.g., 1, a, a1, a-1), placing them according to the chosen scheme. **Right** Distribution of medical image modalities, number of subfigures per compound figure, and caption length statistics within OPEN-PMC-18M. The average caption contains 165.82 tokens, with a max of 7352 and almost 19.48% of captions had more than 256 tokens.

diversity and mimicking real-world figure complexity. Figure 2) illustrates the full synthetic data pipeline.

**Image Decomposition Model Training and Evaluation.** We train a DAB-DETR model on the 500,000 synthetic compound figures and validate its performance on a similarly created holdout set of 20,000 images. Source subfigures are drawn from well-known benchmark datasets such as ROCO (Pelka et al., 2018), SICAP (Ángel E. Esteban et al., 2019), HAM10000 (Tschandl et al., 2018), PathMNIST and RetinaMNIST from MedMNIST (Yang et al., 2021, 2023), PAD-UFES-20 (Pacheco et al., 2020), and PlotQA (Methani et al., 2020) as listed in Table 1. To ensure balanced representation, each modality-specific dataset contributes approximately 16.7% of the total examples, with the remaining 16.7% comprising mixed-modality compound figures. This configuration promotes both visual diversity and generalization across biomedical imaging types. Training is performed over 40 epochs using a batch size of 64 and an initial learning rate of 1e-5. We evaluate performance on both our synthetic validation set and the ImageCLEF 2016 compound figure separation benchmark (Kalpathy-Cramer et al., 2014; García Seco de Herrera et al., 2016). Our model outperforms the model trained on MedICaT only on both evaluation sets as shown by Table 2. Figure 2 showcases examples from the ImageCLEF 2016 dataset and from a subset of PMC-6M, illustrating accurate detection of distinct subfigures across diverse panel layouts and content types.

Table 1: Datasets used for synthetic subfigure generation, categorized by modality and split.

| Split | Radiology | Histopathology | Dermatology | Retina | Plots |
|---|---|---|---|---|---|
| **Train** | ROCO | SICAP | HAM10000 | RetinaMNIST | PlotQA |
| **Sample Size** | 65422 | 18783 | 10015 | 1080 | 60000 |
| **Validation** | ROCO (test) | PathMNIST | PAD-UFES-20 | RetinaMNIST (val) | PlotQA (val) |
| **Sample Size** | 8176 | 10004 | 2298 | 120 | 10000 |

Table 2: Performance comparison on two datasets using mAP and F1 metrics.

| Model | Synthetic Validation | | ImageCLEF 2016 | |
|---|---|---|---|---|
| | **mAP (%)** | **F1 (%)** | **mAP (%)** | **F1 (%)** |
| Previous model (MedICaT) | 33.22 | 73.18 | 28.20 | 64.85 |
| Our model (DAB-DETR) | **98.58** | **99.96** | **36.88** | **73.55** |

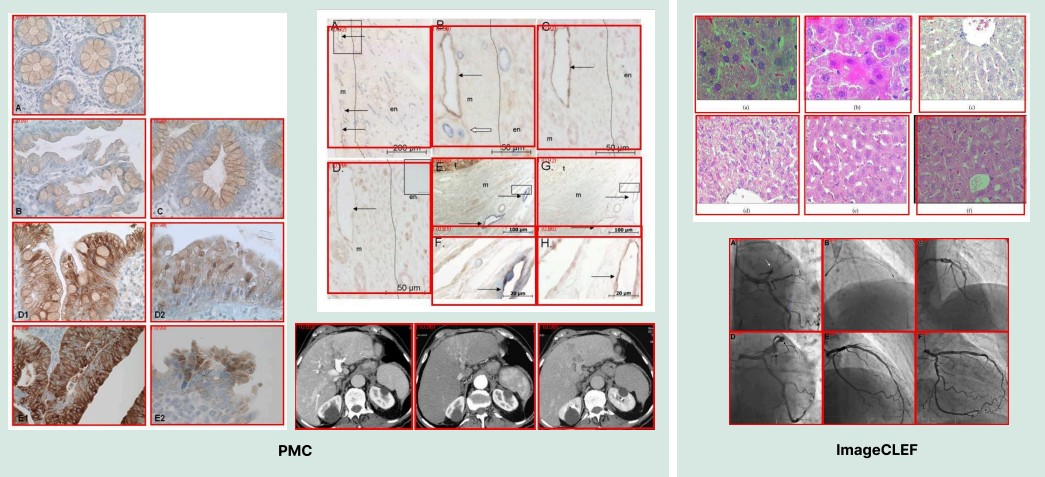

Figure 2: Qualitative results of subfigure detection using our DAB-DETR model. **Left** Real-world biomedical compound figures from PMC articles in PMC-6M (also BIOMEDICA). **Right** Examples from the ImageCLEF 2016 benchmark. The model accurately localizes and separates distinct subfigures, including heterogeneous panels and non-uniform layouts.

## 3.3 Curating OPEN-PMC-18M

Decomposing the compound images of PMC-6M using our DAB-DETR model yields an initial dataset of approximately 32 million single-panel images representing a wide range of clinical (e.g., radiology, pathology, microarray) and non-clinical (e.g., plots) images. For each figure, we simply pair the caption of the source compound figure to create the image-caption pair.

**Filtering Pipeline** To further refine the raw collection of 32 million image-caption pairs, we apply an additional layer of filtering by reviewing metadata fields to only keep subfigures whose original compound figure was labeled by either Clinical Image or Microscopy, which yields a dataset of 26 million pairs. Subsequently, we employ a ResNet-101 model (Lin et al., 2023a) to assess each image and infer its medical relevance. This filtering process further reduces the dataset to 18 million high-quality image-caption pairs.

**Dataset Statistics** We summarize the key characteristics of OPEN-PMC-18M below:

- **Image Modalities:** The dataset includes subfigures from three primary biomedical image modalities, as illustrated in Figure 1: radiology scans (e.g., CT, MRI, X-ray) comprising 18% of the dataset, pathology and microscopy images accounting for 73%, and visible light photography (VLP) representing 8%.

- **Caption Length:** Captions vary in length and complexity. The average caption contains 165.8 tokens. The maximum length is 7352 and almost 19.48% of captions have more than 256 tokens.

## 4 Experiments

### 4.1 Encoder Pretraining

As a first step, we train separate encoders for image and text modalities by aligning their representations using a vanilla contrastive loss. Let $\varphi$ denote an image encoder and $\psi$ denote a text encoder that maps images and text to a common representation space, respectively. Given a batch of training samples $B = \{(x_i, t_i)\}_{i=1}^N$, where $x_i$ and $t_i$ denote the $i^{\text{th}}$ image and text instances respectively, the InfoNCE loss (Oord et al., 2018) is optimized by minimizing the distance between the representations of an image and its corresponding text, $(\varphi(x_i), \psi(t_i))$, while maximizing the distance between

unrelated image-text representation pairs, $(\varphi(x_i), \psi(t_j))$, $i \neq j$:

$$\ell_{\text{con}}(x_i, t_i; B) = -\left( \log \frac{\exp(\langle\varphi(x_i), \psi(t_i)\rangle/\tau)}{\sum_{k=1}^{N} \exp(\langle\varphi(x_i), \psi(\boldsymbol{t_k})\rangle/\tau)} + \log \frac{\exp(\langle\varphi(x_i), \psi(t_i)\rangle/\tau)}{\sum_{k=1}^{N} \exp(\langle\varphi(\boldsymbol{x_k}), \psi(t_i)\rangle/\tau)} \right), \quad (1)$$

where $\langle\cdot, \cdot\rangle$ denotes similarity between two vectors (e.g. cosine similarity), and $\tau > 0$ is a temperature parameter. For simplicity of notation, we drop $B$ and denote the loss for $(x, t)$ by $\ell_{\text{con}}(x, t)$. Multimodal contrastive learning trains encoders $\varphi$ and $\psi$ by minimizing Eq. 1 over the pairs in $B$:

$$\ell_{\text{multimodal}} = \min_{\varphi, \psi} \quad \mathbb{E}_B \left[ \frac{1}{N} \sum_{i=1}^{N} \ell_{\text{con}}(x_i, t_i) \right]. \quad (2)$$

## 4.2 Evaluation Setup

To systematically assess the impact of dataset scale and curation quality, we perform evaluations along both dimensions. Our models are trained under a unified architecture and training protocol to ensure controlled evaluation. For models without accessible training data, we instead use publicly released checkpoints obtained from HuggingFace. For the text encoder, we use PubMedBERT (Gu et al., 2020), and for the vision encoder, we adopt a ViT-B/16 transformer (Dosovitskiy et al., 2020) pretrained on ImageNet. The encoders are trained for 64 epochs with batch size of 2048. The best-performing checkpoints for each encoder are selected based on validation retrieval performance. The training was performed using 8 NVIDIA A100 GPUs and completed in five days. We conducted our experiments using the mmlearn multimodal learning framework, available at https://github.com/VectorInstitute/mmlearn/tree/main.

For assessing the role of quality, particularly subfigure-level extraction, we train a baseline model on the 6 million compound figure-caption pairs of PMC-6M, where each compound image is used in its original form without panel separation (section 3.2). We also include publicly available checkpoints from other models trained on PMC-15M (Zhang et al., 2023) and BIOMEDICA (Lozano et al., 2025). For BIOMEDICA, we use the checkpoint referred to as BMC-CLIP$_{\text{CF}}$ in Lozano et al. (2025), which is trained on a filtered subset of the full dataset. This subset retains content labeled under clinical and scientific imaging, immunoassays, illustrative diagrams, chemical structures, maps, tools and materials, and hand-drawn or screen-based visuals, while explicitly excluding tables and charts. The model is trained for 36 epochs. For PMC-15M, we use the checkpoint trained on 15 million image-caption pairs, referred to as BioMedCLIP in Zhang et al. (2023). All external checkpoints were obtained from their official HuggingFace repositories and are evaluated using our standardized downstream protocols.

To further ensure consistency, we independently reproduce the PMC-OA dataset (Lin et al., 2023b) and train encoders using the same architecture and hyperparameters as those used for OPEN-PMC-18M and PMC-6M. Throughout the paper, all encoder variants are referenced by the name of the dataset on which they are trained, to facilitate transparent comparison. All the details of pretraining and hyperparameters are listed in the supplementary material.

## 4.3 Downstream Tasks

The performance of the encoders is evaluated on external and non-PMC datasets across two primary tasks: retrieval and zero-shot classification. For the retrieval task, we assess both image-to-text (I2T) and text-to-image (T2I) retrieval across three benchmark datasets representative of distinct medical imaging modalities: Quilt (Ikezogwo et al., 2024) (microscopy), MIMIC-CXR (Johnson et al., 2019) (radiology), and DeepEyeNet (Huang et al., 2021) (VLP). To evaluate robustness in retrieval, we follow established protocols from Liu et al. (2024) by applying a suite of low-level visual perturbations, including brightness adjustment, spatial shift, rotation, horizontal flip, and zoom, directly to the test images. To assess the statistical significance of robustness differences, we employ the Wilcoxon signed-rank test, a non-parametric method for paired comparisons (Wilcoxon, 1945). We consider a p-value less than 0.01 as statistically significant. For classification, we evaluate models using both zero-shot and linear probing protocols across a diverse set of tasks: five in radiology, eight in microscopy, and six in VLP. We use our trained vision and text encoders to encode the image and question, respectively.

## 4.4 Cross-Modal Retrieval and Robustness

Table 3 summarizes the performance of various VLMs on cross-modal retrieval tasks across three benchmark datasets: MIMIC-CXR, Quilt, and DeepEyeNet. We report Recall@200 (other Recall metrics are listed in the supplementary material) for both image-to-text and text-to-image retrieval, with the final column showing the Average Recall (AR) aggregated across all tasks. Models trained on OPEN-PMC-18M and even PMC-6M (compound figures) consistently outperform PMC-15M and BIOMEDICA, across all three tasks and retrieval directions. Among them, PMC-6M achieves the highest AR of 21.22, while OPEN-PMC-18M sets a new state-of-the-art with an AR of 21.64. This represents 31% relative gain in average retrieval performance over PMC-15M.

Robustness, quantified as the ratio between retrieval performance under perturbations (explained in section 4.2) and performance on the original data is presented in Figure 3 (Right). Models trained on OPEN-PMC-18M consistently achieve higher robustness scores relative to baseline models, reflecting improved performance stability under input perturbations in addition to superior retrieval performances. We observe statistically significant differences ($p < 0.01$) on Quilt and DeepEyeNet as shown in Figure 3 (Left). These findings are particularly relevant to our focus on subfigure extraction and the potential for improved robustness in imaging modalities that exhibit high visual and semantic heterogeneity.

Table 3: Retrieval performance (Recall@200) of all models trained on paired image-caption pairs in the medical domain. The last column, Average Recall (AR), aggregates the results across all tasks. Highest performance values are in bold, second-best are underlined. PMC-6M refers to a baseline model trained on a filtered subset of the BIOMEDICA dataset, using compound figures in their original form without subfigure decomposition. The BIOMEDICA model retrieved from Hugging Face is trained on a filtered subset of the full dataset, as described in their original paper.

| Model | Image-to-Text | | | Text-to-Image | | | AR |
| --- | --- | --- | --- | --- | --- | --- | --- |
| | MIMIC | Quilt | DeepEyeNet | MIMIC | Quilt | DeepEyeNet | |
| PMC-OA | 0.139 | 0.142 | 0.152 | 0.152 | 0.149 | 0.157 | 0.148 |
| OPEN-PMC | 0.17 | 0.166 | 0.183 | 0.189 | 0.162 | 0.147 | 0.17 |
| BioMedCLIP | 0.185 | 0.165 | 0.162 | 0.162 | 0.185 | 0.146 | 0.167 |
| BIOMEDICA | 0.076 | 0.169 | 0.155 | 0.093 | 0.195 | 0.145 | 0.139 |
| PMC-6M | **0.25** | 0.203 | 0.172 | **0.257** | 0.22 | 0.170 | 0.212 |
| OPEN-PMC-18M | 0.226 | **0.211** | **0.196** | 0.239 | **0.233** | **0.193** | **0.216** |

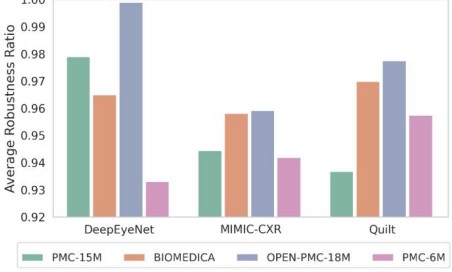

| Model | PMC-6M | BIOMEDICA | PMC-15M |
| --- | --- | --- | --- |
| DeepEyeNet | 0.0014 | 0.0073 | $p > 0.01$ |
| Quilt | 0.0032 | $p > 0.01$ | 0.0001 |
| MIMIC-CXR | $p > 0.01$ | $p > 0.01$ | $p > 0.01$ |

Figure 3: **Left** Average robustness ratio across three retrieval benchmarks, defined as the ratio of retrieval performance under visual perturbations to that on original (unperturbed) data. **Right** Paired statistical comparisons (Wilcoxon signed-rank test) between OPEN-PMC-18M and each baseline model. Results show statistically significant improvements ($p < 0.01$) on DeepEyeNet and Quilt for at least one baseline comparison, while differences on MIMIC-CXR are not statistically significant across any of the baselines.

## 4.5 Zero-shot Classification

Model comparisons for zero-shot classification are presented in Table 4, and linear probing results are provided in the supplementary material. Results are grouped and averaged by modality. Models trained on OPEN-PMC-18M consistently achieve the highest average performance across modalities,

demonstrating superior transferability relative to all other evaluated models. Across the full set of 18 classification tasks spanning radiology, microscopy, and VLP , OPEN-PMC-18M ranks first in 6 tasks and second in 2. A similar trend is observed in the linear probing results, where OPEN-PMC-18M also achieves the highest average performance across modalities.

Table 4: Zero-shot classification F1-scores across diverse medical datasets for different models. For details on model training configurations and dataset sources, refer to the retrieval results table and its caption (Table 3).

| Model | **Radiology** | | | | | |
| | PneumoniaMNIST+ | BreastMNIST+ | OrganAMNIST+ | OrganCMNIST+ | OrganSMNIST+ | Average |
|---|---|---|---|---|---|---|
| PMC-OA | 50.94 | 52.36 | 19.70 | 14.79 | 16.99 | 30.95 |
| OPEN-PMC | 50.13 | **59.65** | **27.95** | **23.23** | **20.03** | 36.19 |
| BioMedCLIP | 60.13 | 33.76 | 19.40 | 14.12 | 16.00 | 28.62 |
| BIOMEDICA | 38.46 | 56.66 | 19.25 | 17.13 | 16.33 | 29.56 |
| PMC-6M | 68.81 | 26.87 | 23.48 | 14.68 | 17.57 | 30.28 |
| OPEN-PMC-18M | **86.18** | 50.36 | 18.75 | 14.33 | 13.65 | **36.65** |

| Model | **Visible Light Photography** | | | | | |
| | PAD-UFES-20 | Skin Cancer | PathMNIST+ | DermaMNIST+ | OCTMNIST+ | RetinaMNIST+ | Average |
|---|---|---|---|---|---|---|---|
| PMC-OA | 17.18 | 13.30 | 56.03 | 14.29 | **50.74** | **27.22** | 29.79 |
| OPEN-PMC | 21.11 | 13.56 | 49.16 | 14.60 | 45.27 | 26.12 | 28.30 |
| BioMedCLIP | 24.41 | 13.62 | 42.27 | 14.07 | 11.87 | 20.82 | 21.17 |
| BIOMEDICA | **40.57** | 17.20 | 49.10 | **21.89** | 10.00 | 18.53 | 26.21 |
| PMC-6M | 33.04 | 16.56 | 52.17 | 17.52 | 46.91 | 22.81 | 31.50 |
| OPEN-PMC-18M | 24.38 | **18.28** | **60.75** | 17.01 | 46.28 | 23.15 | **31.64** |

| Model | **Microscopy** | | | | | | | |
| | Sicap | PCam | NCT-CRC-HE | LC-Lung | LC-Colon | BACH | BloodMNIST+ | TissueMNIST+ | Average |
|---|---|---|---|---|---|---|---|---|---|
| PMC-OA | 32.80 | 70.65 | 43.95 | 56.04 | **91.05** | 33.75 | 5.57 | **7.17** | 42.62 |
| OPEN-PMC | 20.71 | 38.96 | 42.88 | 63.97 | 88.38 | 41.31 | 10.73 | 6.08 | 39.12 |
| BIOMEDICA | 31.80 | 62.17 | 48.98 | 70.93 | 84.43 | 39.83 | 4.37 | 4.31 | 43.35 |
| BioMedCLIP | **41.53** | 72.57 | 49.46 | 76.63 | 86.54 | 23.88 | 6.83 | 3.86 | 45.16 |
| PMC-6M | 22.89 | 68.05 | 55.28 | **86.86** | 78.41 | 52.58 | 3.72 | 3.05 | 46.35 |
| OPEN-PMC-18M | 16.29 | 69.55 | **64.42** | 86.01 | 71.94 | **67.94** | **28.42** | 3.74 | **51.03** |

## 4.6 Representations Analysis

To explore differences in the structure of learned image representations, we project the embedding spaces of three benchmark sets, each constructed by combining datasets used for retrieval and zero-shot classification across radiology, microscopy, and visible light photography (VLP), into two dimensions using t-SNE (Figure 4). The radiology benchmark includes MIMIC-CXR and other related zero-shot classification tasks, totaling approximately 41,000 samples. The microscopy and VLP benchmarks contain approximately 20,000 and 6,000 samples, respectively. To quantify differences between the embedding distributions, we compute the Maximum Mean Discrepancy (MMD) Gretton et al. (2012). Given a dataset $X$ (e.g., all radiology samples), we extract embeddings $\phi(X)$ and $\psi(X)$ using vision encoders $\phi$ and $\psi$ trained on OPEN-PMC-18M and PMC-6M, respectively. To assess whether the differences between these distributions are statistically significant, we perform a permutation test by randomly reassigning samples and recomputing MMD over 100 iterations to generate an empirical null distribution.

Visual inspection of the embeddings reveals distinct representational structures between the two models. This distinction is particularly evident in microscopy and VLP, where the latent spaces of the two models are more clearly differentiated. In contrast, radiology embeddings appear more intermixed, with less visual separation between the models' representation spaces. Nonetheless, the MMD analysis confirms that the observed differences are statistically significant across all modalities. For the aggregated radiology dataset, the observed MMD is 0.0214 (null range: 0.0186–0.0214; $p = 0.005$). For the aggregated microscopy dataset, the observed MMD is 0.0212 (null range: 0.0188–0.0212; $p < 0.001$). For the VLP dataset, the observed MMD is again 0.0214 (null range: 0.0186–0.0214; $p = 0.007$). These results indicate that models trained on subfigure-level data yield significantly different representation spaces compared to those trained on compound figures.

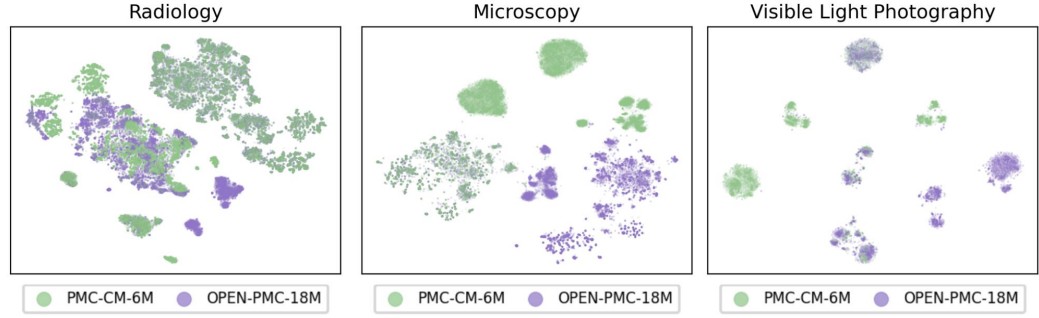

Figure 4: t-SNE visualizations of models embeddings trained on OPEN-PMC-18M and PMC-6M on three imaging modalities, illustrating the structure and separation of the learned representation spaces. MMD analysis reveals statistically significant differences in embedding distributions across all imaging modalities.

## 5 Limitations and Open Challenges in Biomedical Vision–Language Representation Learning

Our findings suggest that in the context of VLM representation learning, data quality and dataset scale should be viewed as complementary axes in building effective and robust biomedical VLMs. Subfigure extraction, used here as a means to improve alignment quality demonstrates clear benefits, particularly in visually heterogeneous domains such as microscopy and visible light photography, as shown in Figure 2. Radiology, however, exhibits more limited gains. These observations raise the importance of modality-aware pretraining strategies, where both model architectures and data curation pipelines are adapted to the unique characteristics of each imaging modality. While our results highlight promising trends, we note that additional analysis is required, particularly in radiology, across a broader and more diverse set of downstream tasks. Such evaluation will help clarify when and where subfigure extraction yields the greatest benefit. Given the strong performance and robustness of encoders trained on OPEN-PMC-18M, future work includes exploring their integration with large language model decoders for downstream tasks that require generative reasoning over visual inputs, such as medical report generation and visual question answering.

We recognize that scaling and curating large biomedical datasets brings challenges that extend beyond improving model performance. To support transparency and reproducibility, we release all dataset filtering criteria, subfigure detection models, and training pipelines. However, interpretability remains an open challenge in VLMs and particularly in the biomedical domain. Although our models are not intended for clinical deployment, they could be fine-tuned or adapted for various clinical application. However, without rigorous validation and careful consideration of clinical safety, such use poses serious risks. Furthermore, our datasets, sourced from open-access repositories such as PMC, may reflect underlying biases tied to specific institutions, imaging protocols, or publication norms. These factors can influence model behavior in subtle ways, limiting generalizability, especially when applied to underrepresented populations or distinct clinical settings.

## 6 Conclusion

In this paper we addressed a critical gap in the design of high-fidelity multimodal medical datasets, aiming to advance robust and generalizable representation learning. We evaluated the effectiveness and robustness of subfigure extraction. We introduced OPEN-PMC-18M, one of the largest and highest quality image-caption pairs to date. Models trained on OPEN-PMC-18M consistently outperform existing benchmarks across radiology, microscopy, and visible light photography. These findings lay the groundwork for more generalizable medical VLMs and better aligned with the complex realities of biomedical data.

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
