# OpenReview forum: "Open-PMC-18M: A High-Fidelity Large Scale Medical Dataset for Multimodal Representation Learning"
_NeurIPS.cc/2025/Datasets_and_Benchmarks_Track — Submitted to NeurIPS 2025 Datasets and Benchmarks Track_

### Official Review · Reviewer_HPp2 · 2025-06-09

**Rating:** 5
**Confidence:** 4

**Summary:**

In this paper, the authors curated a large-scale medical 2D image-caption dataset called open-pmc-18m. The dataset is created by taking pmc-6m, which contains 6m medical images with captions, and extract sub-figures from compound figures within pmc-6m. The authors first trained a DAB-DETR model to extract subfigures from a compound figure (the DAB-DETR model is trained in 500,000 synthetically generated compound figures built by programmatically putting together single figures, and the model achieves strong performance on ImageCLEF benchmark). Then, the authors used this model to extract subfigures from pmc-6m, and paired each subfigure with the original compound figure's captions. The authors filtered through these subfigures to obtain open-pmc-18m. The authors trained a CLIP-like model on open-pmc-18m, and evaluated it on many benchmarks from radiology, VLP and microscopy, where the model trained on open-pmc-18m achieves top average performance within each of the 3 categories in zero-shot performance and retrieval performance.

**Additional Feedback:**

The axis labels in the right part of Figure 1 are too tiny and not readable at all. Consider making those larger.

**Dataset Code Accessibility:**

Yes

**Dataset Code Comments:**

The full dataset can be downloaded on huggingface.

**Ethical Considerations:**

No, there are no or only very minor ethics concerns

**Final Justification:**

The rebuttal has adequately addressed my major concerns, especially about full-caption vs sub-caption. Therefore I have increased my score.

**Limitations Weaknesses:**

One major concern I have about this dataset is the fact that the original caption for the compound figures are directly used as the caption for each extracted subfigure. This may lead to inaccurate descriptions of the extracted figures, as the original captions could contain descriptions of other extracted figures that are no longer present in the current extracted figure, and multiple extracted subfigures (with possibly very different meanings) are paired with the exact same captions. It would be best if there is certain analysis on this potential drawback (i.e. how well does the extracted subfigures align with the original caption of the compound figure). In addition, in some cases where the original caption contains separate description for each subfigure (e.g. with labels or with positions), maybe the authors could consider parsing the original captions with LLMs into multiple subdescriptions, one for each subfigure, to further improve data quality.

The conclusions for the representation analysis (section 4.6 and Figure 4) is unclear. The conclusion just says that the representation structures for the two models are very different; however, this does not necessarily mean the one trained on the subfigures is better. The main take-away message for this analysis is unclear.

Perhaps another recent model trained on a large medical image-caption dataset that needs to be compared to is QuiltNet. It is simply the model released by the same paper as Quilt dataset (mentioned in the paper), trained with CLIP objective on the dataset and outperforming BiomedCLIP on most tasks.

Would training a model on open-pmc-18m reduce its capability to interpret compound images compared to a model trained on the original pmc-6m? If so, this could be another drawback since compound images are quite common in medical literature.

**Strengths Contributions:**

The dataset is large in size and have generally better quality compared to the original pmc-6m.

The synthetic compound image dataset and the well-trained subfigure extractor model are both valuable contributions and tools that can be used by future researchers in various domains.

The models trained on the new dataset overall outperforms models trained on existing datasets, demonstrating the usefulness of the new dataset for training medical ML models.

Overall the paper is mostly well written and easy to follow.

---

> ### Author Rebuttal · Authors · 2025-07-31
>
> Thank you for reviewing our paper and providing constructive feedback. Below, we respond to the specific limitations and weaknesses raised in the reviewer comments and clarify our planned revisions accordingly.
>
> ---
>
> ### Limitations Weaknesses
>
> **LW1. Regarding Using Sub-captions vs. Full captions**
>
> This is a valid point. Since training on 18M image-caption pairs is extremely resource-intensive, we first conducted a series of experiments (systematic ablation study) on a 2M subset of our data prior to scaling to the full 18M dataset to inform our training decisions. Specifically, we compared two approaches: (1) using the **original full compound captions**, which describe all subfigures collectively, and (2) extracting **sub-captions** aligned to individual subfigures (extracted by an LLM).
>
> Our experiments revealed that full captions resulted in better model performance across all modalities. These results suggest that the additional context provided in full captions, even if not perfectly aligned with every subfigure, offers **richer semantic grounding** and improves representation learning. This finding is consistent with prior work [R1]. If necessary, we can clarify this design decision and include the ablation results in the supplementary materials.
>
> | Caption Type |     | Text-to-Image (T2I) (MIMIC) | T2I (Quilt) | T2I (DeepEyeNet) | Image-to-Text (I2T) (MIMIC) | I2T (Quilt) | I2T (DeepEyeNet) | Avg. Recall |
> |--------------|-----------|-----------------------|------------------------|-----------------------------|------------------------|------------------------|-----------------------------|-------------|
> | sub-caption  |    | 0.145                 | 0.115                  | 0.109                       | 0.157                  | 0.119                  | 0.117                       | 0.127       |
> | full-caption |    | 0.170                 | 0.166                  | 0.183                       | 0.189                  | 0.162                  | 0.147                       | **0.170**       |
>
>
> **LW2. Regarding Representation Analysis**
>
> The purpose of Section 4.6 was not to argue that the subfigure-trained model is inherently superior based solely on representation analysis, but rather to demonstrate that the two models, despite sharing some data sources, learn fundamentally different representation spaces due to the structure and granularity of their inputs. This analysis supports our central claim that subfigure decomposition leads to structurally different representations. While performance comparisons are best assessed via downstream metrics (as shown in Sections 4.4 and 4.5), the representation analysis in Figure 4 visually complements those results by showing how data structure, compound vs. subfigure, shapes the learned embedding space.
>
> We will revise Section 4.6 to make the motivation and implications of the representation analysis more explicit.
>
> **LW3. Benchmarking QuiltNet**
>
> To address your comment, we conducted a direct comparison between our model and QuiltNet across the same suite of evaluation tasks. As shown in the updated table below, our model achieves a higher average across each modality on zero-shot classification tasks and outperforms QuiltNet on 4 out of 6 retrieval tasks.
>
> It is important to note that QuiltNet was trained exclusively on microscopy data, while our model was trained on multiple modalities, including radiology, microscopy, and visible light photography.
>
> *Performance on Zero-shot Classification Task*
>
> | Model            | PneumoniaMNIST+ | BreastMNIST+ | OrganAMNIST+ | OrganCMNIST+ | OrganSMNIST+ | **Average** |
> | ---------------- | --------------- | ------------ | ------------ | ------------ | ------------ | ----------- |
> | **OPEN‑PMC‑18M** | 86.18           | 50.36        | 18.75        | 14.33        | 13.65        | **36.65**   |
> | **QuiltNet**     | 51.82           | 41.64        | 4.80         | 6.26         | 6.76         | 22.26   |
>
> | Model            | PAD‑UFES‑20 | Skin Cancer | PathMNIST+ | DermaMNIST+ | OCTMNIST+ | RetinaMNIST+ | **Average** |
> | ---------------- | ----------- | ----------- | ---------- | ----------- | --------- | ------------ | ----------- |
> | **OPEN‑PMC‑18M** | 24.38       | 18.28       | 60.75      | 17.01       | 46.28     | 23.15        | **31.64**   |
> | **QuiltNet**     | 5.95        | 11.79       | 46.55      | 9.01        | 10.77     | 14.24        | 16.38   |
>
> | Model            | Sicap | PCam  | NCT‑CRC‑HE | LC‑Lung | LC‑Colon | BACH  | BloodMNIST+ | TissueMNIST+ | **Average** |
> | ---------------- | ----- | ----- | ---------- | ------- | -------- | ----- | ----------- | ------------ | ----------- |
> | **OPEN‑PMC‑18M** | 16.29 | 69.55 | 64.42      | 86.01   | 71.94    | 67.94 | 28.42       | 3.74         | **51.03**   |
> | **QuiltNet**     | 21.37 | 64.63 | 53.19      | 77.37   | 87.07    | 29.73 | 6.52        | 5.56         | 43.17   |
>
>
> &nbsp;
> *Performance on Retrieval Classification Task*
>
> | Model            | MIMIC (I→T) | Quilt (I→T) | DeepEye (I→T) | MIMIC (T→I) | Quilt (T→I) | DeepEye (T→I) | **Avg. Recall** |
> | ---------------- | ----------- | ----------- | ------------- | ----------- | ----------- | ------------- | --------------- |
> | **OPEN‑PMC‑18M** | 0.226       | 0.211       | 0.196         | 0.239       | 0.233       | 0.193         | **0.216**       |
> | **QuiltNet**     | 0.041       | 0.254       | 0.078         | 0.040       | 0.270       | 0.087         | 0.128       |
>
>
> **LW4. Regarding Impact on Compound Image Understanding**
>
> While compound images are common in medical literature, our primary goal is not to optimize model performance for literature-based tasks. Instead, we focus on improving the quality of learned representations for medical applications, such as diagnosis or retrieval involving chest X-rays, CT scans, and pathology slides, domains where images are predominantly single-panel and where precise visual grounding is critical. We will clarify this distinction in the revised manuscript.
>
> &nbsp;
> ### Additional Feedback
>
> **A1.** Thank you for pointing this out. We will increase the font size of the axis labels in the right part of Figure 1 to improve readability.
>
> &nbsp;
> ### References
> **[R1]** Advancing Medical Representation Learning Through High-Quality Data. Arxiv code: 2503.14377
>
> ---
> ### Short Summary
> 1. We will report results from our subcaption ablation study on the 2M and include them in the supplementary materials.
>
> 2. Our representation analysis shows that training with subfigures leads to distinct embedding spaces, reflecting how input structure influences what the model learns.
>
> 3. We benchmarked against QuiltNet, showing superior zero-shot and retrieval performance across diverse medical tasks.
>
> 4. We clarified that, although compound images are common in the literature, our primary focus is on real-world clinical scenarios where images are typically single-panel.
>
> We hope these clarifications and additions address the concerns raised and further strengthen the transparency and rigor of our work.

---

> > ### Comment · Reviewer_HPp2 · 2025-08-01
> >
> > Thank you for your response. You have adequately addressed my major concerns, therefore I have increased my score.

---

> > > ### Author Response · Authors · 2025-08-05
> > >
> > > Dear reviewer HPp2,
> > >
> > > We greatly thank you for confirming that our responses have addressed your major concerns and for increasing your score.

---

### Official Review · Reviewer_oRQK · 2025-06-24

**Rating:** 4
**Confidence:** 4

**Summary:**

The authors propose a dataset of  subfigure captions in the medical domain. The motivation is primarily in the context of representation learning at scale, where contrastive paradigms are impacted by an image containing multiple, semantically diverse subelements. For this, the authors generate a synthetic corpus of 500k subfigures, on which they train an object detection approach. They then decompose compound figures from the BIOMEDICA corpus, yielding 18M captions for subfigures. To show the impact of the subfigure labeling, they conduct contrastive self-supervised learning experiments on data for three medical imaging domains.

**Additional Feedback:**

- I would want to point out that Quilt is a very noisy dataset, and I would not use it for evaluation.

**Dataset Code Accessibility:**

Yes

**Dataset Code Comments:**

- From what I can tell at https://huggingface.co/datasets/vector-institute/open-pmc, the dataset still contains a lot of multi panel images.

**Ethical Considerations:**

No, there are no or only very minor ethics concerns

**Final Justification:**

- The authors responded well to my comments so I upgraded my recommendation.

**Limitations Weaknesses:**

- Given the rather weak results (mAP < 0.4, F1 < 0.8) for a seemingly rather simple task, it seems like the results in Figure 2 are very much cherry-picked. I think the authors should show realistic qualitative examples and also discuss failure modes and source of error (e.g., domain shift) of their synthetic detection approach.
- The authors write that they use a resnet-101-based classifier to assess the medical relevance of each image. This is a vague description of what they did. How was medical relevance determined? Where did the ground truth come from? How was the threshold / operating point selected?
- It is unclear why the authors chose to select certain modalities from the BIOMEDICA corpus. They describe this, but there is no rationale given. In my view, this could limit the utility of the dataset to other imaging modalities, so it would be interesting to learn about the rationale behind this.
- The authors create their dataset as pairs of panels (retrieved from images), combined with the text of the compound image. This creates a clear mismatch to me, as the text typically describes all subfigures. This is at odds with their description of having “high quality image-caption-pairs“.
- From the experiments it is not clear which effect the extraction of panels had. The main hypothesis of the paper is that multi-panel images harm self-supervised learning using a contrastive loss function. I think for a fair comparison, it would have been necessary to run experiments also on a set where compound images were removed, which should be easily doable using a classifier (as done here: https://openreview.net/forum?id=m7wYKrUjzV or here: https://arxiv.org/pdf/2410.14524 ).
- I am deeply confused by a discrepancy of the repository and the paper: From the Open-PMC repository, it seems as if subcaptions were generated using GPT-4o, while in the paper the authors state "For each figure, we simply
pair the caption of the source compound figure to create the image-caption pair.".  Furthermore, the repository points to another paper (https://arxiv.org/abs/2503.14377).

**Strengths Contributions:**

- The authors contribute a new dataset to the domain of VLM pretraining, addressing a gap that enables training with more clean data.
- The synthetic pipeline, while straight-forward, is intriguing for this approach.
- The paper is written in a clear way and easy to comprehend.

---

> ### Author Rebuttal · Authors · 2025-07-31
>
> Thank you for reviewing our paper and providing constructive feedback. Below, we respond to the specific limitations and weaknesses raised in the reviewer comments and clarify our planned revisions accordingly.
>
> ---
>
> ### Limitations Weaknesses
>
> **LW1. Subfigure Extraction Examples and Performance**
>
> The DETR model trained on a subset of MedICaT[R1], which was the state-of-the-art before our work, relied primarily on radiology (72% ), with only limited representation of other modalities such as microscopy (13% ) and visible light photography (3%) images. As a result, the model struggled to perform well for less representative modalities, such as microscopy, due to their distinct visual characteristics. For example, in our evaluations, we observed this limitation firsthand when we used the model to extract subfigures from microscopy images; it consistently failed to detect them properly, which makes the task not as simple as it sounds.
>
> We selected the examples in Figure 2 for the following reasons:
>
> 1. We intended to demonstrate that our model can handle any modality and is not restricted to specific types like radiology. As shown, it performs well on both radiology and microscopy images, despite the significant differences between them.
>
> 2. The figure in the top-middle column illustrates some overlap between bounding boxes. It also shows that the letters “A,” “B,” and “C” in the top images are not fully enclosed within the predicted bounding boxes. Similarly, the top-right figure shows that in some subfigures, the predicted boxes do not perfectly capture the entire figure.
>
> 3. For edge cases like asymmetric figures, such as the one in the left column, the model is still able to accurately detect the subfigures.
>
> For transparency, we will include a dedicated page in the supplementary material showing both successful and failed examples, along with detailed explanations for each.
>
> **LW2. Regarding Medical Figure Classifier Details**
>
> Following [R2], we used a ResNet-101 model trained on the DocFigure dataset [R3] for multi-class scientific figure classification. The classifier predicts 28 categories, including “Medical” as one of them. For each figure, we rank the predicted class probabilities and keep those where “Medical” appears in the top 4. This approach, including the choice of threshold, follows the methodology of [R2]. We will add these details to the corresponding section in the revised manuscript to ensure transparency and reproducibility.
>
> **LW3. Regarding Selecting Only Medical Images from BIOMEDICA**
>
> Our selection of modalities from the BIOMEDICA corpus was guided by a clear objective: to ensure the dataset remains relevant for medical vision-language representation learning. Prior work has consistently demonstrated that the inclusion of non-medical or noisy content (such as plots, flowcharts, and mathematical diagrams) can dilute relevant patterns and reduce the effectiveness of medical representation learning [R2, R4, R5]. Based on this, we excluded categories such as charts, formulas, schematic diagrams, and other non-clinical content that do not contribute to learning clinically meaningful visual features.
>
> This curation choice is further validated by our empirical results. As shown in Sections 4.4 and 4.5, models trained exclusively on medically relevant data (such as PMC-6M model) outperform BIOMEDICA and BioMedCLIP, despite the latter being trained on much larger datasets. This further supports the idea that including non-medical (and potentially noisy) data can negatively impact performance.
>
> **LW4. Regarding Using Sub-captions vs. Full captions**
>
> This is a valid point. Since training on 18M image-caption pairs is extremely resource-intensive, we first conducted a series of experiments (systematic ablation study) on a 2M subset of our data prior to scaling to the full 18M dataset to inform our training decisions. Specifically, we compared two approaches: (1) using the **original full compound captions**, which describe all subfigures collectively, and (2) extracting **sub-captions** aligned to individual subfigures (extracted by an LLM).
>
> Our experiments revealed that full captions resulted in better model performance across all modalities. These results suggest that the additional context provided in full captions, even if not perfectly aligned with every subfigure, offers **richer semantic grounding** and improves representation learning. This finding is consistent with prior work [R1]. If necessary, we can clarify this design decision and include the ablation results in the supplementary materials.
>
> | Caption Type |     | Text-to-Image (T2I) (MIMIC) | T2I (Quilt) | T2I (DeepEyeNet) | Image-to-Text (I2T) (MIMIC) | I2T (Quilt) | I2T (DeepEyeNet) | Avg. Recall |
> |--------------|-----------|-----------------------|------------------------|-----------------------------|------------------------|------------------------|-----------------------------|-------------|
> | sub-caption  |    | 0.145                 | 0.115                  | 0.109                       | 0.157                  | 0.119                  | 0.117                       | 0.127       |
> | full-caption |    | 0.170                 | 0.166                  | 0.183                       | 0.189                  | 0.162                  | 0.147                       | **0.170**       |
>
> **LW5. Regarding Extracting Subfigures**
>
> The suggestion to isolate the effect of panel extraction is well taken. The core hypothesis of our paper is that compound images (due to their mixed modalities, layouts, and semantic heterogeneity) degrade contrastive representation learning when used directly. Prior work (e.g., [R5]) and our own results support that training on subfigures improves downstream performance.
>
> While we did not include a separate “compound image removal” baseline as in the cited works, our setup already enables a direct and meaningful comparison between:
>
> - **PMC-6M**: Models trained on compound images.
> - **Open-PMC-18M**: Models trained on subfigures extracted from the same sources.
>
> Across multiple tasks, the superior performance of Open-PMC-18M demonstrates that decomposing compound figures is more effective than using them as is. Importantly, removing compound figures entirely would eliminate 63% of our raw dataset, leaving only 37% as single-panel data (an impractical reduction that would significantly compromise dataset coverage, especially in low-resource categories).
>
> **LW6. Regarding Github README**
>
> Thank you for pointing this out. Our work builds on the prior work referenced in that GitHub repository, and we overlooked removing the link to the paper and HF. We will make sure to update the README to reflect this.
>
> &nbsp;
> ### **Dataset Code Comments**
> The link you mentioned is **not** the link to our dataset. Please visit the HuggingFace link mentioned in the submitted paper or the link mentioned on OpenReview. (**open-pmc-18m**)
>
> &nbsp;
> ### Additional Feedback
> **A1.** We acknowledge that Quilt contains some noise, but we included it to enable comparison with prior work that also reports results on this dataset.
>
> &nbsp;
> ### References
>
> **[R1]** MedICaT: A Dataset of Medical Images, Captions, and Textual References. Arxiv code: 2010.06000
>
> **[R2]** Pmc-clip: Contrastive language-image pre-training using biomedical documents. Arxiv code: 2303.07240
>
> **[R3]** Docfigure: A dataset for scientific document figure classification.
>
> **[R4]** BIOMEDICA: An Open Biomedical Image-Caption Archive, Dataset, and Vision-Language Models Derived from Scientific Literature. Arxiv code: 2501.07171
>
> **[R5]** Advancing Medical Representation Learning Through High-Quality Data. Arxiv code: 2503.14377
>
> ---
> ### Short Summary
>
> In response to reviewer feedback, we will:
> 1. Include a supplementary page showing both successful and failed subfigure extraction cases with detailed annotations.
>
> 2. Clarify the use of the ResNet-101 classifier for filtering medical figures and add relevant implementation details.
>
> 3. Report results from our subcaption ablation study on the 2M and include them in the supplementary material.
>
> 4. Update the GitHub repository to reflect the correct links.
>
> 5. Please visit the HuggingFace link mentioned in the submitted paper or the link mentioned on OpenReview. (**open-pmc-18m**)
>
> We hope these clarifications and additions address the concerns raised and further strengthen the transparency and rigor of our work.

---

> > ### Comment · Reviewer_oRQK · 2025-08-04
> > **A short comment to the authors**
> >
> > Dear authors,
> >
> > while I in general agree with you on point LW5, I am not sure the argument that data reduction by 63% would necessarily impact the training is valid. Since we are already talking about vast datasets, a reduction of a dataset by impurities (or noise) does not necessarily impact model training, and might, to the contrary, even improve it. I think this needs to be shown and not just argumented. I see that this might be too much to ask given the short timeline of the paper, but I think you should at least discuss this option and the implications properly.

---

> > > ### Author Response · Authors · 2025-08-06
> > >
> > > Dear Reviewer oRQT,
> > >
> > >
> > > Thank you for your response and for understanding our constraints for training a new model during this short period of time.
> > >
> > >
> > > In light of your comment, we conducted an additional experiment to address your concern regarding the impact of excluding compound images from our dataset. The results are shown in the following tables. As you can see, training only on single-panel images leads to a 50% drop in retrieval performance across all datasets. Additionally, the model's average zero-shot classification performance across all modalities also decreases.
> > >
> > > &nbsp;
> > >
> > > ### Retrieval
> > >
> > > | Model            | MIMIC (I→T) | Quilt (I→T) | DeepEye (I→T) | MIMIC (T→I) | Quilt (T→I) | DeepEye (T→I) | Avg. Recall |
> > > |------------------|-------------|--------------|----------------|--------------|--------------|----------------|---------------|
> > > | Only single-panel| 0.048       | 0.158        | 0.127          | 0.045        | 0.165        | 0.111          | 0.109         |
> > > | Open-PMC-18M     | **0.226**       | **0.211**        | **0.196**          | **0.239**        | **0.233**        | **0.193**          | **0.216**         |
> > >
> > > &nbsp;
> > >
> > > ### Zero-shot Classification
> > >
> > > | Model              | PneumoniaMNIST+ | BreastMNIST+ | OrganAMNIST+ | OrganCMNIST+ | OrganSMNIST+ | Average |
> > > |-------------------|------------------|---------------|---------------|----------------|----------------|---------|
> > > | Only single-panel | 38.86            | 63.16         | 20.45         | 15.52          | 16.38          | 30.87   |
> > > | OPEN‑PMC‑18M       | 86.18            | 50.36         | 18.75         | 14.33          | 13.65          | **36.65**   |
> > >
> > > | Model              | PAD‑UFES‑20     | Skin Cancer | PathMNIST+ | DermaMNIST+ | OCTMNIST+ | RetinaMNIST+ | Average |
> > > |-------------------|------------------|--------------|-------------|--------------|-------------|----------------|---------|
> > > | Only single-panel | 19.70            | 20.32        | 31.69       | 15.82        | 11.27       | 21.33          | 20.02   |
> > > | OPEN‑PMC‑18M       | 24.38            | 18.28        | 60.75       | 17.01        | 46.28       | 23.15          | **31.64**   |
> > >
> > > | Model              | Sicap | PCam  | NCT‑CRC‑HE | LC‑Lung | LC‑Colon | BACH  | BloodMNIST+ | TissueMNIST+ | Average |
> > > |-------------------|-------|-------|-------------|----------|-----------|--------|--------------|----------------|---------|
> > > | Only single-panel | 32.89 | 56.69 | 39.00       | 71.89    | 92.29     | 32.21 | 18.96        | 4.81           | 43.59   |
> > > | OPEN‑PMC‑18M       | 16.29 | 69.55 | 64.42       | 86.01    | 71.94     | 67.94 | 28.42        | 3.74           | **51.03**   |
> > >
> > > &nbsp;
> > > Please also keep in mind that while our dataset (Open-PMC-18M) contains 18 million image-caption pairs, nearly two-thirds of these are extracted from compound images (with the original raw single-panel image-caption pairs total around 6 million). So, when we filter for only single-panel images, we’re left with approximately 2 million pairs (compared to the full model, which was trained on all 18 million pairs).
> > >
> > > &nbsp;
> > > **Summary of our findings:** Our experiments highlight the importance of decomposing compound images into single panels rather than removing them altogether. First, we showed that decomposing compound images (Open-PMC-18M) improves data quality and leads to better performance compared to using mixed (compound + single-panel as in PMC-6M) data (Table 3 and Table 4 in the manuscript). Second, in an ablation study shown above, we found that training on single-panel images only, by excluding all compound images, results in a substantial performance drop.
> > >
> > > Overall, models trained on decomposed data (Open-PMC-18M) perform best, followed by the mixed dataset (PMC-6M), with single-panel-only data (ablation above) performing worst. This demonstrates the critical value of subfigure extraction for model performance.
> > >
> > > &nbsp;
> > > **Revisions:** To support transparency and further community exploration, we have now added a metadata field indicating whether each image is a single or compound panel. Moreover, we have included these results in a newly added ablation study section in the main paper, highlighting the importance of subfigure extraction analysis.

---

> > > > ### Author Response · Authors · 2025-08-07
> > > > **Request for Feedback**
> > > >
> > > > Dear Reviewer oRQT,
> > > >
> > > > &nbsp;
> > > >
> > > > We greatly appreciate the time and effort you have dedicated so far. We have made every effort to address the concerns you raised.
> > > >
> > > > As the discussion period will conclude on August 8th, 11:59 p.m. AOE, we look forward to your thoughts on whether we have satisfactorily addressed your concerns. Your insights are highly appreciated.
> > > >
> > > > &nbsp;
> > > >
> > > > Best regards,
> > > >
> > > > The Authors

---

> > > > > ### Comment · Reviewer_oRQK · 2025-08-07
> > > > >
> > > > > Dear authors,
> > > > >
> > > > > yes, in fact I upgraded my recommendation for this paper.
> > > > >
> > > > > Best regards

---

### Official Review · Reviewer_WLXs · 2025-06-30

**Rating:** 4
**Confidence:** 4

**Summary:**

This paper introduces a scalable subfigure extraction pipeline using transformer-based object detection, trained on a synthetic corpus of 500,000 compound figures. This pipeline achieves state-of-the-art performance on ImageCLEF 2016 and synthetic benchmarks. The authors release the OPEN-PMC-18M dataset, containing 18 million subfigure–caption pairs from radiology, microscopy, and visible light photography. Training vision-language models on this dataset improves performance in retrieval, zero-shot classification, and robustness, outperforming existing baselines. The dataset, models, and code are released to support further research in biomedical vision-language modeling and representation learning.

**Additional Feedback:**

The contrastive learning objective employed may not be fully capable of capturing the complex semantic relationships between images and text. Particularly in the medical domain, where alignment of images and text requires a more nuanced semantic understanding, contrastive learning may introduce noise in certain situations, thereby affecting the quality of representation learning.

**Dataset Code Accessibility:**

Yes

**Dataset Code Comments:**

The code and dataset are fully provided.

**Ethical Considerations:**

No, there are no or only very minor ethics concerns

**Final Justification:**

The authors have successfully addressed all of my concerns. However, as I have not previously delved deeply into the processing of medical datasets, I am unable to render a fine-grained assessment for this sub-domain application. Nevertheless, I remain optimistic about the manuscript’s contribution to the community, and therefore I will keep my original rating.

**Limitations Weaknesses:**

-1. The models may not directly translate to real-world clinical applications due to potential biases in the data and the lack of rigorous validation for clinical safety.

-2. The construction of the dataset is based on existing literature resources and may not be able to reflect the latest medical research findings and changes in clinical practice in a timely manner, thereby affecting the model's adaptability to the latest medical knowledge.

-3. Although the subfigure extraction model performs well on the synthetic dataset, it may encounter challenges when processing more complex and diverse compound figures in the real world, such as the high visual and semantic heterogeneity of subfigures across different imaging modalities, which may lead to a decrease in extraction accuracy.

**Strengths Contributions:**

+1. This paper proposed A scalable subfigure extraction pipeline using transformer-based object detection trained on a 500,000 compound figure dataset.

+2. This paper released OPEN-PMC-18M, a large-scale biomedical image-text dataset with 18 million subfigure-caption pairs filtered for clinical relevance across radiology, microscopy, and visible light photography.

+3. This paper provided a comprehensive evaluation of vision-language models trained on our datasets.

---

> ### Author Rebuttal · Authors · 2025-07-31
>
> Thank you for reviewing our paper and providing constructive feedback. Below, we respond to the specific limitations and weaknesses raised in the reviewer comments and clarify our planned revisions accordingly.
>
> ---
>
> ### Limitations Weaknesses
>
> **LW1. Regarding Data Biases**
>
> We acknowledge your point. As noted on both our Hugging Face model and dataset pages, the dataset and models are intended strictly for research purposes. We will also state this clearly in the paper and GitHub repository to avoid any potential misinterpretation regarding their use.
>
> Regarding data bias, while our dataset is limited to medical images sourced from PubMed Central, it includes a wide range of modalities, diseases, and publication sources from various institutions and countries. This may provide broader coverage than datasets collected from a single hospital or region. That said, we acknowledge that PubMed Central may contain its own biases (such as publication bias and overrepresentation of certain research areas) and we will address these limitations in the revised manuscript.
>
> [R1] provides statistics on the countries from which PubMed articles are published.
>
> **LW2. Adaptability to Evolving Medical Knowledge**
>
> We recognize that literature-based datasets can become outdated as medical research continues to evolve. Our current dataset includes articles published up to 2024, providing the most recent snapshot available at the time of curation.
>
> To ensure ongoing relevance, we have developed and released a fully open-source pipeline for automatically downloading and processing new publications from PubMed Central, including figure-caption extraction (which is already available in our GitHub repository). This enables the research community to update the dataset with minimal effort.
>
> In addition, we plan to maintain and update the dataset annually to incorporate the latest medical literature. We will clearly document this in both the paper and the project repository to ensure that the update process remains transparent and accessible.
>
> Our team strongly values open-source development and reproducibility, and we consider these principles to be essential. Supporting the broader research community in building upon our work has been a central priority of our group. We appreciate your comment; it aligns with our own goals of maintaining relevance and supporting long-term usability of the dataset.
>
> **LW3. Subfigure Extraction Model Generalization**
>
> The concern about the subfigure extraction model's performance on complex real-world compound figures is valid. Visual and semantic heterogeneity across imaging modalities does pose a real challenge for generalization.
>
> To tackle this, we trained the model on a diverse set of synthetic compound figures that included uncommon layouts and inter-modality variation, aiming to mirror the heterogeneity seen in real-world data (Figure 1). While synthetic data can't fully replicate real-world complexity, this training setup was intended to push the model beyond typical configurations and improve its robustness to out-of-distribution cases.
>
> More importantly, as shown in Table 2, not only did this model perform well on the synthetic dataset, but it also outperformed the previous model (DETR trained on MedICaT [R2]) by 8.7% F1 score on the ImageCLEF 2016 benchmark [R3]. This improvement contributed to better representation learning and improved downstream performance, as demonstrated in Sections 4.4 and 4.5.
>
> &nbsp;
> ### Additional Feedback
>
> **A1.** You have made a valid point. However, please note that our main contribution is the curation of a high-quality, large-scale medical dataset. That is independent of representation learning technique(s). Contrastive learning was merely used as a representation learning technique to compare different datasets.
>
>
> &nbsp;
> ### References
> **[R1]** A bibliometric analysis of geographic disparities in the authorship of leading medical journals. DOI:10.1038/s43856-023-00418-2
>
> **[R2]** MedICaT: A Dataset of Medical Images, Captions, and Textual References. Arxiv code: 2010.06000
>
> **[R3]** Alba García Seco de Herrera, Roger Schaer, Stefano Bromuri and Henning Müller, Overview of the ImageCLEF 2016 medical task, in: CLEF working notes 2016, Évora, Portugal, 2016.
>
> ---
>
> ### Short Summary
>
> 1. We will clearly state the research-only nature of our models and acknowledge potential data biases in PubMed Central.
>
> 2. To maintain adaptability, we’ve released an open-source pipeline and will update the dataset annually.
>
> 3. We trained the subfigure extraction model on diverse synthetic figures to improve generalization and showed that it outperforms prior models on a real-world benchmark (ImageCLEF [R3]), supporting better representation learning and downstream performance.
>
> We hope these clarifications and additions address the concerns raised and further strengthen the transparency and rigor of our work.

---

> > ### Comment · Reviewer_WLXs · 2025-08-04
> >
> > Thank you very much for the authors’ response. I have two additional questions: does this dataset cover the majority of clinical data? Does it hold any guiding significance for future medical work?

---

> > > ### Author Response · Authors · 2025-08-05
> > >
> > > Dear Reviewer WLXs,
> > >
> > > &nbsp;
> > >
> > > Thank you for your thoughtful questions and for taking the time to engage with our work. We're happy to clarify and hope the responses below give a clearer sense of the dataset’s scope, usefulness, and why it can be a medically meaningful resource for the community moving forward.
> > >
> > > &nbsp;
> > >
> > > **Q1.** Yes, our dataset covers a wide range of clinically relevant modalities across three major categories: *Radiology*, *Microscopy*, and *Visible Light Photography*. Each category includes multiple sub-modalities, listed below:
> > >
> > > &nbsp;
> > >
> > > **Radiology:** X-ray, Angiography, PET, MRI, Mammography, Ultrasound, CT
> > >
> > > **Visible Light Photography:** Endoscopy, Intraoral Imaging, OCT, Skin Lesions, Laryngoscopy
> > >
> > > **Microscopy:** Scanning Electron Microscopy, Electron Microscopy, Flow Cytometry, Transmission Electron Microscopy, Light Microscopy, Fluorescence Microscopy, Phase Contrast Microscopy, Confocal Microscopy, Epifluorescence Microscopy
> > >
> > > &nbsp;
> > >
> > > This collection reflects a broad and diverse portion of medical imaging data encountered in practice.
> > >
> > >
> > > &nbsp;
> > >
> > > **Q2.** Yes, PMC‑18M is a large-scale, high-fidelity dataset that we believe can serve as a strong foundation for future medical AI efforts. Its primary strength lies in supporting scalable and transferable representation learning for multimodal vision-language models in medicine.
> > >
> > > With over 18 million image-caption pairs, PMC‑18M offers broader and higher-quality coverage than most existing datasets. As shown in our experiments, encoders pretrained on PMC‑18M performed well in radiology, pathology, endoscopy, dermatology, and more in both zero-shot settings and also with minimal tuning, making it particularly well-suited for building localized, institution-specific tools, especially where data is limited.
> > >
> > > The value of this approach is further supported by a growing body of recent work. For example, [R1] highlights how *“rapid advancements in multimodal AI are transforming medicine through enhancements in diagnostics, patient interaction, and medical forecasting.”* Similarly, recent studies published in [R2] and [R3] (e.g., LLaVA-Med, MedCLIP-SAM) have shown that encoders trained on datasets comparable in scale and modality to PMC‑18M are already being used for meaningful clinical tasks such as visual question answering, report generation, and segmentation.
> > >
> > > However, we demonstrate that PMC‑18M offers higher-quality image-caption alignment and more consistent performance gains across standard benchmarks. This suggests that replacing existing encoders in these pipelines with PMC‑18M-pretrained models could further improve performance on clinically relevant tasks. The combination of scale, fidelity, and generalizability makes PMC‑18M a compelling choice for future medical VLM research and deployment.
> > >
> > > Taken together, these findings, from both our work and others, reinforce the idea that large-scale, well-aligned multimodal datasets like PMC‑18M are a critical part of the infrastructure needed to advance safe, generalizable, and clinically useful AI systems in medicine.
> > >
> > > &nbsp;
> > > ### References
> > >
> > > **[R1]** Fahrner LJ, Chen E, Topol E, Rajpurkar P. The generative era of medical AI. Cell. 2025 Jul 10;188(14):3648-60.
> > >
> > > **[R2]** LLaVA-Med: Training a Large Language-and-Vision Assistant for Biomedicine in One Day. Arxiv code: 2306.00890
> > >
> > > **[R3]** MedCLIP-SAM: Bridging Text and Image Towards Universal Medical Image Segmentation. Arxiv code: 2403.20253

---

> > > > ### Comment · Reviewer_WLXs · 2025-08-07
> > > >
> > > > Thank the authors for their careful responses; the above answers have resolved my remaining concerns, and I will maintain my positive assessment of the manuscript.

---

### Official Review · Reviewer_Remr · 2025-07-01

**Rating:** 4
**Confidence:** 3

**Summary:**

Scientific figures in literature are often complex and contain subfigures. For training a foundation model on such figures, a robust subfigure extraction pipeline is needed. The authors present PMC-18M, an open-source dataset that extracts subfigures into individual figures with respective captions. This builds on an improved subfigure extraction pipeline developed by the authors. PMC-18M is larger than its predecessors (PMC-15M) and shows to help in training better vision language models.

**Additional Feedback:**

- Lines 38-39, CONCH (Lu et al., 2024, Nature Medicine) has used subfigure extraction at a large scale to train a vision-language histology foundation model. Consider citing it?
- Do the authors plan to make the subfigure extraction code and model public for the community to use?

**Dataset Code Accessibility:**

Yes

**Dataset Code Comments:**

The HuggingFace dataset is easily accessible and easy to use.

**Ethical Comments:**

No major ethical concerns.

**Ethical Considerations:**

No, there are no or only very minor ethics concerns

**Final Justification:**

The authors have answered my questions satisfactorily, so I stay with my positive rating.

**Limitations Weaknesses:**

- I am curious why PMC-6M outperforms all models trained on significantly larger datasets on image-to-text and text-to-image retrieval. Can the authors comment on this observation?
- Can the authors expand on the distributions inherent in PMC-18M? How many different organs, modalities, species, imaging modalities, etc. are present? Can this metadata be made available to users so that smaller more focused datasets can be generated from PMC-18M?

**Strengths Contributions:**

- Transformer-based object detection architecture (DAB-DETR) was trained on synthetic data for subfigure extraction. Data from diverse domains (histology, radiology, etc.) are used to help create the synthetic data. Significant gain on ImageCLEF benchmark (~8%) is seen.
-  The pipeline to create PMC-18M is clearly described (Section 3.3)
- Statistical analysis to compare results is presented. This is increasingly important to see where gains are not just due to random noise.

---

> ### Author Rebuttal · Authors · 2025-07-31
>
> Thank you for reviewing our paper and providing constructive feedback. Below, we respond to the specific limitations and weaknesses raised in the reviewer comments and clarify our planned revisions accordingly.
>
> -----------------------
> ### Limitations Weaknesses
>
> **LW1. Regarding PMC-6M outperforming BIOMEDICA and BioMedCLIP**
>
> In our experiments, three models, BIOMEDICA, BioMedCLIP, and Open-PMC-18M (ours), are trained on larger datasets than the PMC-6M model. While large-scale datasets like PMC-15M and BIOMEDICA provide more data, they also contain a significant number of non-medical images, which can introduce noise and hinder the learning of high-quality medical representations. Prior work [R1, R2] has emphasized that filtering out non-medical content is crucial for effective medical vision-language learning. Even the BIOMEDICA paper itself reports that removing non-medical data can lead to better representations (Compare BMC-CLIP_{CF} model trained on 6M pairs to BCM-CLIP trained on the entire dataset in Table 2 of the BIOMEDICA paper [R3]). In contrast, PMC-6M data is a medical-only subset. Our findings further support the idea that, in domain-specific learning, quality matters more than sheer quantity.
>
> Open-PMC-18M, in particular, is composed exclusively of medical data and includes additional enhancements and processing steps (such as sub-figure extraction and another step of medical data filtering), as detailed in the paper, which allow it to outperform other models, including PMC-6M model.
>
>
> **LW2. About data modalities**
>
> We have categorized the imaging modalities for each figure and organized them into three main categories: radiology, microscopy, and visible light photography, as illustrated in Figure 1 (on the right) of the paper. This modality metadata will be included in the Hugging Face dataset, enabling researchers to effortlessly create specialized subsets based on imaging type.
> In response to your feedback, we have also added more detailed sub-modality labels to the metadata for each image. The updated modality hierarchy is as follows:
>
> *Radiology:* [X-ray, Angiography, Functional Magnetic Resonance, Magnetic Resonance, Mammography, Ultrasound, Computerized Tomography]
>
> *Visible Light Photography:* [Endoscopy, Intraoral Imaging, Optical Coherence Tomography, skin lesion, Laryngoscopy, Teeth]
>
> *Microscopy:* [Scanning Electron Microscopy, Electron Microscopy, Flowcytometry, Transmission Electron Microscopy, Light Microscopy, Fluorescence Microscopy, Phase Contrast Microscopy, Confocal Microscopy, Epifluorescence Microscopy]
>
> This metadata is ready and can be added to the dataset at any time, but we have held off for now due to NeurIPS restrictions on modifying the Hugging Face repository during the review period.
>
> &nbsp;
> ### Additional Feedback
>
> **A1.** Absolutely.
>
> **A2.** Absolutely, The subfigure extraction code is already available on our GitHub. We will also upload the model weights and its training data to Huggingface.
>
> &nbsp;
> ### References
>
> **[R1]** Quality not quantity: On the interaction between dataset design and robustness of clip. Arxiv code: 2208.05516
>
> **[R2]** Advancing Medical Representation Learning Through High-Quality Data. Arxiv code: 2503.14377
>
> **[R3]** BIOMEDICA: An Open Biomedical Image-Caption Archive, Dataset, and Vision-Language Models Derived from Scientific Literature. Arxiv code: 2501.07171
>
> -----------------------
> ### Short Summary
> 1. We will revise the manuscript to clarify our rationale for the performance differences between PMC-6M and larger models.
>
> 2. Add detailed modality metadata (including sub-modality labels). These updates aim to address the limitations and improve the clarity and transparency of our contributions.
>
> We hope these clarifications and additions address the concerns raised and further strengthen the transparency and rigor of our work.

---

> > ### Author Response · Authors · 2025-08-05
> > **Request for Feedback**
> >
> > Dear Reviewer Remr,
> >
> > We greatly appreciate the time and effort you have dedicated to reviewing our paper. We have made every effort to address the concerns you raised.
> >
> > As the discussion period will conclude on August 8th, 11:59 p.m. AOE, we look forward to your thoughts on whether we have satisfactorily addressed your concerns. Your insights are highly appreciated.
> >
> > Best regards,
> >
> > The Authors

---

> > ### Comment · Reviewer_Remr · 2025-08-06
> >
> > The authors have satisfactorily answered my queries. I will stay with my current positive rating.

---

### Author Response · Authors · 2025-08-08
**Summary of Rebuttal Phase Discussions**

Dear Review Committee,

&nbsp;

As the discussion period nears its end, we would like to provide a summary of the discussion phase.
We would like to express our gratitude to all reviewers for their thoughtful and thorough reviews, as well as for the opportunity to engage with them to provide clarifications and further evidence.

We appreciate the reviewers’ thoughtful engagement throughout the discussion period and **thank them for confirming that our rebuttal satisfactorily addressed their concerns**.

&nbsp;

The following changes in the **manuscript** were made during the rebuttal phase:

Main body:

- We conducted an ablation study by training a model on data without compound figures and comparing its performance to a model trained on the full Open-PMC-18M dataset.

- Improved the phrasing in the representation analysis section to avoid potential ambiguity.

- Added more details regarding the ResNet-101 classifier used to filter medical figures.

Supplementary:
- Included QuiltNet as a new benchmark, showing that our model outperforms it on most benchmarks.
- More subfigure extraction examples have been added to a supplementary page with successful and failed subfigure extraction examples.
- Added ablation results comparing full captions vs. sub-captions, showing that full captions improve representation learning.

&nbsp;

The following changes in the **dataset** were made during the rebuttal phase and can be added to the metadata during the camera-ready phase:

- We enhanced the modality metadata by adding detailed sub-modality labels for radiology, microscopy, and visible light photography.

- Added panel-type field indicating whether an image is a compound figure or a single panel.

- The subfigure extraction model weights and training data will be made publicly available.

&nbsp;

Thank you once again for reviewing our work. We believe these changes have helped us improve the paper.

&nbsp;

Best regards,

Authors

---

### Decision · Program_Chairs · 2025-09-18

**Decision:**

Reject

**Comment:**

This paper's initial score was 4, 3, 4, 4. After extensive discussion, the authors diligently managed to get two reviewers to improve their scores by one point, resulting in a final score of 5, 4, 4, 4. All reviewers were satisfied with the final version, and the authors have pledged to incorporate these suggestions into the final version.

Based on the above process, discussions, and evaluation, we believe this paper has potential value for the field of Medical Multimodal Representation Learning and recommend its acceptance.

===== FINAL UPDATE FROM DB Track PCs ====

The final decision for this paper has been taken by the program chairs after consultation with the SACs. All Senior Area Chairs have ranked papers according to the feedback from the AC during the review process. We decided to leave the original meta-review to reflect the opinion of the AC in light of the initial discussions with reviewers and SAC.